# Unfavorable outcomes to second-line tuberculosis therapy among HIV-infected versus HIV-uninfected patients in sub-Saharan Africa: A systematic review and meta-analysis

Dumessa Edessa[1]*, Mekonnen Sisay[2], Yadeta Dessie[3]

1 Department of Clinical Pharmacy, School of Pharmacy, College of Health and Medical Sciences, Haramaya University, Oromia, Ethiopia, 2 Department of Pharmacology and Toxicology, School of Pharmacy, College of Health and Medical Sciences, Haramaya University, Oromia, Ethiopia, 3 School of Public Health, College of Health and Medical Sciences, Haramaya University, Oromia, Ethiopia

* jaarraa444@yahoo.com

**Data Availability Statement:** All relevant data are within the manuscript and its Supporting Information files.

## Abstract

### Background

Drug resistance is a key obstacle to the global target set to end tuberculosis by 2030. Clinical complexities in drug-resistant tuberculosis and HIV-infection co-management could worsen outcomes of second-line anti-tuberculosis drugs. A comprehensive estimate for risks of unsuccessful outcomes to second-line tuberculosis therapy in HIV-infected versus HIV-uninfected patients is mandatory to address such aspects in segments of the target set. Therefore, this meta-analysis was aimed to estimate the pooled risk ratios of unfavorable outcomes to second-line tuberculosis therapy between HIV-infected and HIV-uninfected patients in sub-Saharan Africa.

### Methods

We conducted a literature search from PubMed/MEDLINE, EMBASE, SCOPUS and Google Scholar. We screened the retrieved records by titles and abstracts. Finally, we assessed eligibility and quality of full-text articles for the records retained by employing appraisal checklist of the Joanna Briggs Institute. We analyzed the data extracted from the included studies by using Review Manager Software, version 5.3 and presented our findings in forest and funnel plots. Protocol for this study was registered on PROSPERO (ID: CRD42020160473).

### Results

A total of 19 studies with 1,766 from 4,481 HIV-infected and 1,164 from 3,820 HIV-uninfected patients had unfavorable outcomes. The risk ratios we estimated between HIV-infected and HIV-uninfected drug-resistant tuberculosis patients were 1.18 (95% CI: 1.07–1.30; $I^2$ = 48%; P = 0.01) for the overall unfavorable outcome; 1.50 (95% CI: 1.30–1.74) for

**Funding:** We have received no funding for this work.

**Competing interests:** The authors have declared that no competing interests exist.

death; 0.66 (95% CI: 0.38–1.13) for treatment failure; and 0.82 (95% CI: 0.74–0.92) for loss from treatment. Variable increased risks of unfavorable outcomes estimated for subgroups with significance in mixed-age patients (RR: 1.22; 95% CI: 1.10–1.36) and eastern region of sub-Saharan Africa (RR: 1.47; 95% CI: 1.23–1.75).

## Conclusions

We found a higher risk of unfavorable treatment outcome in drug-resistant tuberculosis patients with death highly worsening in HIV-infected than in those HIV-uninfected patients. The risks for the unfavorable outcomes were significantly higher in mixed-age patients and in the eastern region of sub-Saharan Africa. Therefore, special strategies that reduce the risks of death should be discovered and implemented for HIV and drug-resistant tuberculosis co-infected patients on second-line tuberculosis therapy with optimal integration of the two programs in the eastern region of sub-Saharan Africa.

## Introduction

Tuberculosis (TB) is one of the top 10 causes of death and the leading cause of infectious disease-related mortality [1, 2]. In 2018, there were an estimated 10 million incident TB cases worldwide with about 1.5 million deaths [3]. More than 95% of cases and deaths related to TB infection were occurred in developing countries [4]. The World Health Organization's (WHO) regional report in 2016 indicated a quarter (i.e., 2.5 million) of new cases and 417,000 deaths related to TB disease burden in Africa only [5]. Drug-resistance is a key obstacle to global efforts to end TB infection [6]. Remarkably, drug-resistant TB (DR-TB) including multidrug-resistant TB (MDR-TB) remains a public health crisis and a health security threat [2]. It is further complicated in the presence of the human immunodeficiency virus (HIV) co-infection [6]. For instance, the DR-TB was two times more likely to develop in HIV-infected TB patients than in those HIV-uninfected ones [7]. According to the WHO report in 2013, 3.5% of new cases and 20.5% of previously treated TB cases had MDR-TB [8]. The MDR-TB is an infection that is resistant to rifampicin and isoniazid, both of which are the most powerful drugs in the first-line regimen for TB therapy [4].

The DR-TB can be treated effectively by second-line anti-tuberculosis drugs which are toxic and also require a treatment follow-up for about 2 years [4, 9]. The appropriate regimen involving four or more drugs commonly combined from the core second-line medicines to treat the DR-TB includes group A (levofloxacin, moxifloxacin, gatifloxacin), group B (amikacin, capreomycin, kanamycin, streptomycin), and group C (ethionamide/prothionamide, cycloserine/terizidone, linezolid, clofazimine) plus one drug or none from the add-on agents from group D (pyrazinamide, ethambutol, high-dose isoniazid, delamanid, bedaquiline, p-aminosalicylic acid, imipenem–cilastatin, meropenem, and amoxicillin-clavulanate) [10]. However, the presence of HIV-coinfection influences the successful outcomes to treatment with the second-line anti-tuberculosis medicines; for that, both infections are commonly indicated as cursed duets that exist together and affect the outcomes of each other [11, 12].

High mortality was unfavorably associated with the treatment of DR-TB [13], and such an unsuccessful treatment outcome is more distressing to a community in resource-limited settings such as sub-Saharan Africa (SSA). Again, the rate of unfavorable treatment outcome occurring with the second-line tuberculosis therapy is an aspect which is alarming and

threatening to the global progress towards the end TB strategy targets set by 2030 [9]. The WHO defines the unfavorable treatment outcome for a DR-TB as the sum of the numbers of death, treatment failure, loss to follow-up and unknown outcome identified during the courses of second-line tuberculosis therapy [14]. Prolonged therapy is required for the toxic second-line anti-tuberculosis treatment which is burdensome for patient compliance [13]. This prolonged treatment with the less tolerated second-line anti-tuberculosis treatment regimen might have a lower rate of patient compliance with the treatment that could affect the outcome of therapy [15, 16]. As a result, a lower rate of unfavorable outcome during the DR-TB treatment is a key component that indicates a successful outcome for the epidemic [17]. More importantly, clinical complexity linked to the co-management of DR-TB and HIV-infections, demands comprehensive evidence that helps inform a successful treatment strategy for the second-line anti-tuberculosis drugs in HIV-infection [18]. Accordingly, a comprehensive estimate for risks of unfavorable outcomes to second-line TB therapy among HIV-infected patients compared to the HIV-uninfected ones is mandatory to address such aspects in pillars of the target set. Therefore, this study was aimed to pool the overall risk ratio (RR) for the unfavorable outcome to second-line TB therapy between HIV-infected and HIV-uninfected patients in SSA.

## Methods

### Study protocol

The method of this meta-analysis was reported as per the Preferred Reporting Items for Systematic Review and Meta-Analysis Protocols (PRISMA-P) 2015 statement recommendations [19]. We performed selection of records, screening processes and eligibility evaluations against the predefined inclusion criteria following the PRISMA flow diagram [20]. We also strictly followed the PRISMA checklist during execution of this meta-analysis. Protocol for this meta-analysis was registered on the International Prospective Register of Systematic Reviews (PROSPERO) (ID: CRD42020160473).

### Data search strategy

We conducted systematic searches of databases and legitimate indexing services to identify and include potential records. PubMed/MEDLINE (Ovid), EMBASE (Ovid), and SCOPUS were visited as major sources of data search from December 25, 2019, to February 15, 2020. Besides, we also searched Google Scholar and ResearchGate/directories to retrieve relevant records left unaddressed by the legitimate databases visited. The records identified by ResearchGate/directories were individually saved and linked to Endnote via the Google Scholar. Again, we searched for unpublished studies (grey literature) through the Google Scholar. We considered the unpublished studies to reduce their impacts on the publication bias. The search strategy involved combinations of one or more of the following terms: "second-line*", "drug-resistant", multidrug-resistant (MeSH), tuberculosis (MeSH), "HIV-infection", "loss to follow-up", treatment failure (MeSH), death (MeSH), and Africa, South of the Sahara (MeSH). Moreover, we employed the truncation of search terms and Boolean operators (AND, OR) as appropriate to expand and fine tune the search strategy thereby identify and include more records.

### Inclusion and exclusion criteria

We screened original articles that address treatment outcomes of second-line anti-tuberculosis drugs among HIV-infected patients compared to those HIV-uninfected and reported in the

English language for inclusion in the meta-analysis. As such, we assessed the eligibility of studies reporting unfavorable outcomes (i.e., treatment failure, death, loss from treatment (i.e., either loss to follow-up or treatment default)) to the second-line TB therapy among the HIV-infected versus HIV-uninfected patients with at least one of the outcome in the definition of the unfavorable outcome and conducted in Africa, South of the Sahara. However, we excluded articles with outcomes unrelated to the outcome of interest (i.e., unfavorable outcomes not reported as died, treatment failed or lost from treatment) during the screening and eligibility assessments. Again, we excluded articles that report outcomes of mixed patients from extensively drug-resistant TB and MDR-TB and with no separate outcome report for the MDR-TB during courses of the treatment. Moreover, we excluded articles that fulfilled eligibility evaluation for inclusion but which did not meet the quality requirements.

## Screening and eligibility assessment

First of all, we identified, downloaded, and exported records retrieved through a systematic search of electronic databases, indexing services, and directories with a compatible format to Endnote reference software, version 8.2 (Thomson Reuters, Stamford, CT, USA). Secondly, we identified, registered and removed duplicate records from the shortlisted references by the use of Endnote. Following this, we manually identified and removed duplicates resulting from variation in citation styles of some databases and indexing services. Next, two authors, Dumessa Edessa (DE) and Mekonnen Sisay (MS), independently screened the retained records by their titles and abstracts based on the predefined inclusion criteria. Finally, two authors, DE and Yadeta Dessie (YD), independently collected and evaluated full-texts of the retained articles for eligibility assessments.

## Quality assessment and data extraction

We performed methodological quality assessments of the retained articles for inclusion by using the Joanna Briggs Institute's (JBI) critical appraisal checklist for cohort studies [21]. Accordingly, we employed the appraisal scores of the two authors in consideration of the third author's score in case of appraisal result disagreement between the two authors. Next, we ranked the articles by overall scores of positive responses to questions of the JBI's critical appraisal checklist for methodological qualities. Finally, we included all studies which fulfilled the eligibility requirements and with the overall positive scores higher than 50% in the meta-analysis.

We prepared a data abstraction format in Microsoft excel sheet to extract all relevant data about study characteristics (name of the first author, year of publication, study setting/country, study design, category of study participants (children, adolescents, adults, mixed-age patients), sample size, months of follow-up, and events of interest (number of patients with second-line tuberculosis treatment failure, number of patients died during the treatment and number of patients lost from the treatment follow-up) in both the HIV-infected and HIV-uninfected patients.

## Outcome variables

The pooled RR estimate for the overall unfavorable outcome to second-line tuberculosis therapy among HIV-infected versus HIV-uninfected patients as defined by the WHO [9] was the primary outcome variable. We also conducted subgroup analyses for the overall unfavorable outcome based on categories of study participants and regions of the SSA. The separate pooled RR estimates for deaths, treatment failure and loss from treatment among HIV-infected versus

HIV-uninfected patients treated with second-line tuberculosis therapy were secondary outcome variables we considered.

## Data synthesis and analysis

We generated study identification and outcome for the dichotomous and discrete data type from the included studies for the HIV-infected versus HIV-uninfected DR-TB patients in Review Manager (RevMan) software, version 5.3 to analyze the pooled RR estimate of the overall unfavorable outcome measures and subgroup analyses. We employed the same software to estimate RRs for outcomes including death, treatment failure and loss from treatment. Again, we conducted subgroup analyses for the overall unfavorable outcome by the age category of patients and regions/settings of the SSA. We also conducted subgroup analysis to estimate RR for death outcome by regions of the SSA. Considering the variation in true effect sizes across study subjects, we applied Mantel Haenszel's random-effects method for the analyses at a 95% confidence level. We assessed the variation in study characteristics (heterogeneity) by using $Tau^2$, $chi^2$ and $I^2$ statistics. We also employed RevMan 5.3 for publication bias assessment by the symmetry of funnel plots for standard error of logit RR. We employed forest and funnel plots to present results of our analyses.

## Results

From the systematic electronic search of legitimate databases and indexing services, we retrieved a total of 1134 archives. After removing 280 duplicate records by the use of Endnote and manual screening, we were left with 854 records. Following this, we screened the retained records by their titles and excluded 346 records. Again, we excluded 415 records by screening abstracts. From this 761 total records excluded by screening titles and abstracts, 556 of them were with unrelated outcome of interest; 180 of them were discussion papers; 18 of them were records from outside setting or mixed settings with no separate data for the SSA, and 7 of them were published with non-English languages. Next, we conducted quality and eligibility assessments for 93 full-text articles along with the predefined inclusion and exclusion criteria. Accordingly, we excluded 74 full-text articles with reasons (i.e., 51 of them with irrelevant outcomes; 20 of them with insufficient information and 3 of them with mixed patients from extensively DR-TB and MDR-TB but with no separate outcome reported for MDR-TB treatment). In line with this, we employed JBI's critical appraisal checklist for methodological quality assessment of the retained articles (i.e., see score for each article from S1 Table).

In the end, we included 19 articles with greater than fifty percent of the average positive score for methodological quality assessments and with a report of at least an overall unfavorable outcome or any outcome that constitutes the definition of unfavorable outcome (i.e., death, treatment failure, loss from treatment). PRISMA flow chart illustrating the identification, screening, and eligibility assessment processes is shown in Fig 1 and S2 Table.

## Study characteristics

The 19 studies included in this meta-analysis had 4,481 participants from HIV-infected DR-TB patients compared to the 3,820 participants in the DR-TB infection only. Overall, 1,766 patients out of the 4,481 participants versus 1,164 patients out of the 3,820 participants had unfavorable treatment outcomes among HIV-infected versus HIV-uninfected patients treated with second-line TB therapy. Similarly, 942 versus 455 patients among HIV-infected versus HIV-uninfected patients treated by second-line TB therapy had died from 16 studies included. In line with this, 140 versus 163 patients from among HIV-infected versus HIV-uninfected patients treated with second-line TB therapy had their treatment failed from 10 of

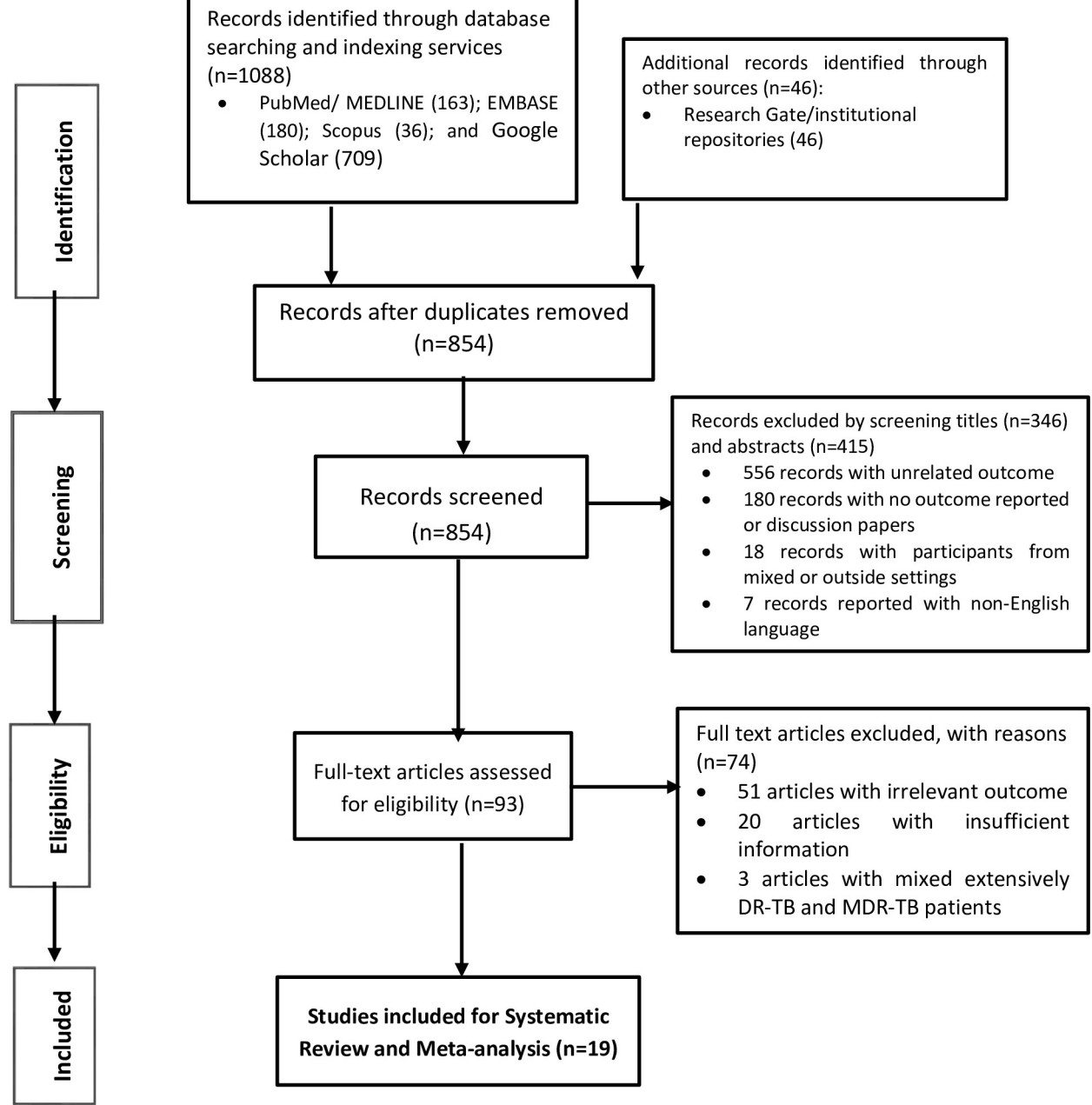

**Fig 1. PRISMA flow diagram illustrating the selection process for the meta-analysis.**

the studies included. Moreover, 641 versus 497 patients from 14 of the included studies among HIV-infected versus HIV-uninfected patients treated with second-line TB therapy had lost from treatment (i.e., treatment defaulted or lost to follow-up). Publication dates of the included studies range from 2011 to 2020. The number of participants in the included studies for HIV-infected patients treated with second-line TB therapy ranges from 11 [22] to 1104 [23], and the number ranges from 40 [24] to 479 [25] for those HIV-uninfected patients treated with second-line TB therapy. The study participants for ten of the included studies [22, 26–34] were mixed-age patients; for 7 of the included studies [23–25, 35–38] were adults; for

one of the included study [39] were adolescents and adults, and for the rest one of the included study [40] were children. Eleven (n = 11) of the included studies were from southern region of SSA [23, 24, 26, 27, 32, 34–37, 39, 40], while 6 [25, 28–31, 33] and 2 [22, 38] of them were from eastern and western regions, respectively (Table 1).

**Table 1. Characteristics of studies describing unfavorable outcomes to second-line tuberculosis therapy among HIV-infected versus HIV-uninfected patients on treatment follow-up in SSA.**

| References | HIV-infected people with DR-TB | | | | | People with DR-TB infection only | | | | | Design | FU duration (months) | Study setting | Age category of patients |
|---|---|---|---|---|---|---|---|---|---|---|---|---|---|---|
| | # FT | # Died | # LFT | # with unfavourable outcome | Sample size | # FT | # Died | # LFT | # with unfavourable outcome | Sample size | | | | |
| Adewumi et al, 2012 [41] | 5 | 16 | 24 | 45 | 94 | 24 | 28 | 52 | 104 | 242 | RFU | 18 | South Africa | Mixed-age |
| Alakaye et al, 2018 [27] | 8 | 64 | 5 | 77 | 265 | – | 19 | 1 | 20 | 78 | RFU | 18 | Lesotho | Mixed-age |
| Alene et al, 2017 [28] | 1 | 9 | 5 | 15 | 51 | 3 | 22 | 22 | 47 | 191 | RFU | 20 | Ethiopia | Mixed-age |
| Brust et al, 2018 [35] | 19 | – | 7 | 26 | 150 | 9 | – | 1 | 10 | 56 | RFU | 32 | South Africa | Adults |
| Cox et al, 2014 [39] | 28 | 66 | 90 | 184 | 351 | 12 | 18 | 56 | 86 | 149 | RFU | 24 | South Africa | Adolescents and adults |
| Farley et al, 2011 [36] | 12 | 101 | 59 | 172 | 287 | 62 | 76 | 99 | 237 | 470 | FU | 11.6 | South Africa | Adults |
| Hall et al, 2017 [40] | 2 | 49 | 14 | 65 | 238 | 1 | 20 | 22 | 43 | 185 | RFU | 24 | South Africa | Children |
| Huerga et al, 2017 [29] | 1 | 9 | 2 | 12 | 35 | – | 13 | 10 | 23 | 110 | RFU | 24 | Kenya | Mixed-age |
| Jikijela et al, 2018 [37] | 11 | 121 | 27 | 159 | 245 | 25 | – | 9 | 39 | 85 | RFU | 24 | South Africa | Adults |
| Ketema et al, 2019 [30] | – | – | – | 30 | 123 | – | – | – | 56 | 385 | RFU | 24 | Ethiopia | Mixed-age |
| Leveri et al, 2019 [31] | | 22 | 8 | 30 | 116 | – | 34 | 15 | 49 | 216 | RFU | 24 | Tanzania | Mixed-age |
| Loveday et al, 2015 [23] | 55 | 180 | 225 | 460 | 1104 | 19 | 43 | 98 | 160 | 376 | FU | 24 | South Africa | Adults |
| Marais et al, 2014 [32] | – | 45 | – | 45 | 203 | – | 11 | – | 11 | 72 | RFU | 24 | South Africa | Mixed-age |
| Mengistu et al, 2019 [33] | – | 13 | – | 13 | 34 | – | 21 | – | 21 | 97 | RFU | 24 | Ethiopia | Mixed-age |
| Meresa et al, 2015 [25] | 4 | 27 | 9 | 40 | 133 | 6 | 58 | 27 | 91 | 479 | RFU | 24 | Ethiopia | Adults |
| Mohr et al, 2015 [34] | – | 101 | 149 | 250 | 539 | – | 22 | 78 | 90 | 218 | RFU | 18 | South Africa | Mixed-age |
| Piubello et al, 2020 [22] | – | – | – | 3 | 11 | – | – | – | 22 | 191 | RFU | 12 | Niger | Mixed-age |
| Satti et al, 2012 [24] | 1 | 29 | – | 30 | 94 | – | 17 | 1 | 18 | 40 | RFU | 22.9 | Lesotho | Adults |
| Shin et al, 2017 [38] | 3 | 90 | 17 | 110 | 408 | 2 | 28 | 7 | 37 | 180 | RFU | 24 | Botswana | Adults |
| **Total** | **140** | **942** | **641** | **1766** | **4481** | **163** | **455** | **497** | **1164** | **3820** | | | | |

#, number; FT, failed treatment; FU, follow-up; LTF, lost from treatment; RFU, retrospective follow-up; SSA, sub-Saharan Africa; '—', no report for the outcome.

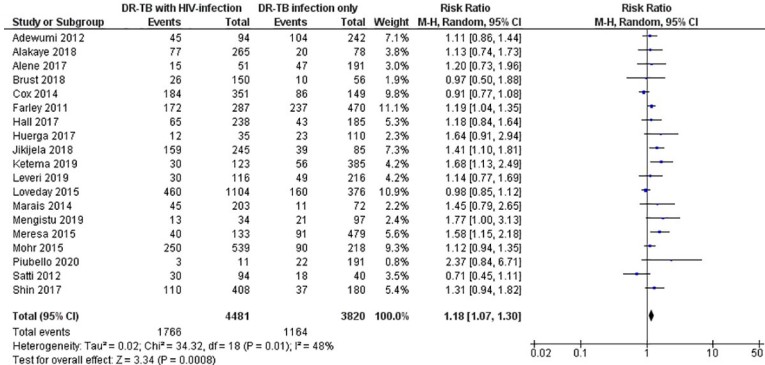

**Fig 2. Forest plot for risk of unfavorable treatment outcome among HIV-infected versus HIV-unifected patients treated with second-line tuberculosis therapy.**

## Pooled risk ratio for unfavorable outcomes

The pooled estimate of RR for the overall unfavorable outcome in HIV-infected versus HIV-uninfected patients treated by second-line anti-tuberculosis drugs was 1.18 (95% CI: 1.07–1.30; Z = 3.34; P = 0.0008; $I^2$ = 48%; P = 0.01). It ranged from 0.71 (95% CI: 0.45–1.11) to 2.37 (95% CI: 0.84–6.71) when outcome measure of the individual studies were considered (Fig 2).

Similarly, the pooled estimate of the RR for death among the HIV-infected versus HIV-uninfected patients treated by second-line anti-tuberculosis treatment was 1.50 (95% CI: 1.30–1.74; Z = 5.55; P<0.00001; $I^2$ = 39%; P = 0.05). The RR estimates for the death outcome in the included studies ranged from 0.73 (95% CI: 0.45–1.16) to 2.18 (95% CI: 1.02–4.65) (Fig 3).

However, we obtained a non-significant RR estimate between HIV-infected patients and the HIV-uninfected counterparts treated by second-line anti-tuberculosis drugs in case of treatment failure (RR: 0.66; 95% CI: 0.38–1.13; Z = 1.52; P = 0.13; $I^2$ = 73%; P = 0.0001). It ranged from 0.15 (95% CI: 0.08–0.30) to 2.40 (95% CI: 0.69–8.38) when outcome measure of the individual studies were considered (Fig 4).

Again, the pooled RR estimate for loss from treatment among the HIV-infected versus HIV-uninfected patients treated by second-line tuberculosis therapy was 0.82 (95% CI: 0.74–0.92; Z = 3.53; P = 0.0004; $I^2$ = 0%; P = 0.47). The estimates for the loss from treatment in the included studies ranged from 0.49 (95% CI: 0.26–0.94) to 2.61 (95% CI: 0.33–20.77) (Fig 5).

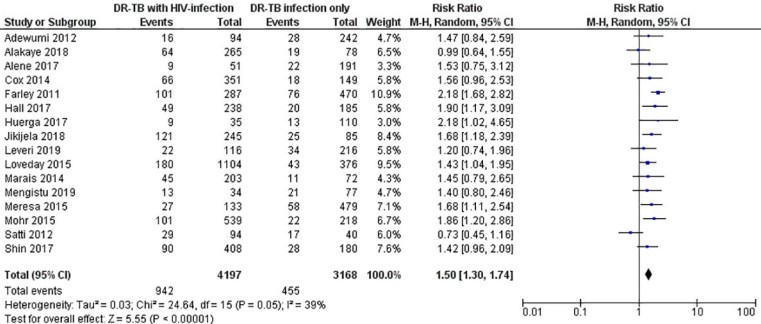

**Fig 3. Forest plot for risk of death among HIV-infected versus HIV-unifected patients treated by second-line tuberculosis therapy.**

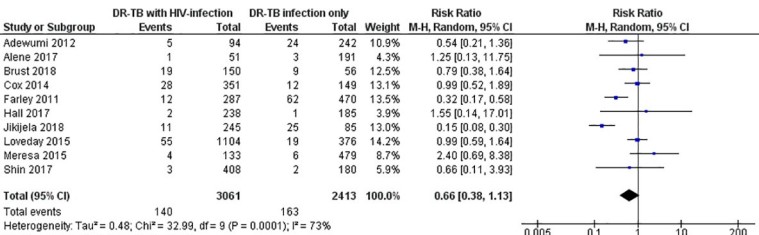

**Fig 4. Forest plot for risk of treatment failure among HIV-infected versus HIV-unifected patients treated by second-line tuberculosis therapy.**

## Sensitivity and subgroup analyses

We executed sensitivity analyses by excluding two outliers [22, 24] and/or more studies [33, 39], but they did not have significant changes on the degree of heterogeneity among the included studies. Following this, we included all studies in the meta-analysis. The studies we considered for the sensitivity analyses as outliers were those studies with RR estimates far smaller and/or greater than the pooled RR estimate for the outcome interest. We also performed subgroup analyses for the overall unfavorable outcome based on age group of patients (adults and adolescents, mixed-age, children) and regions of the SSA in which the studies were carried out (western, eastern, southern) to reduce the degree of heterogeneity in the included studies. As a result, the pooled RR estimates for the unfavorable outcome in the HIV-infected versus HIV-uninfected patients treated by second-line TB therapy were 1.12 (95% CI: 0.96–1.30; $I^2$ = 69%; P = 0.002; Z = 1.44; P = 0.15) in adults and adolescents; 1.23 (95% CI: 1.09–1.37; $I^2$ = 0%; P = 0.49; Z = 3.51; P = 0.0005) in mixed-age patients; 1.18 (95% CI: 0.84–1.64; Z = 0.95; P = 0.34) in children; 1.42 (95% CI: 0.95–2.13; $I^2$ = 12%; P = 0.29; Z = 1.72; P = 0.09) in the western SSA region; 1.47 (95% CI: 1.23–1.75; $I^2$ = 0%; P = 0.63; Z = 4.32; P < 0.0001) in the eastern SSA region and 1.09 (95% CI: 0.98–1.20; $I^2$ = 43%; P = 0.06; Z = 1.64; P = 0.10) in the southern SSA region. Subgroup analysis for the RR estimate of death by regions of the SSA also indicated RR of 1.49 (95% CI: 1.21–1.83; $I^2$ = 60%; P = 0.008; Z = 3.75; P = 0.0002) for southern; 1.52 (95% CI: 1.19–1.93; $I^2$ = 0%; P = 0.73; Z = 3.36; P = 0.0008) for eastern; and 1.42 (95% CI: 0.96–2.09; Z = 1.77; P = 0.08) for western region of SSA (Fig 6A and 6B and S1 Fig).

Similarly, we performed sensitivity analyses by excluding two outliers [24, 36] and/or more studies [27, 29] for the pooled RR estimate of death between the two comparison groups. However, they did not have significant changes in the degree of heterogeneity in the included studies. Besides, we excluded outliers [33, 37] and one or more studies for sensitivity analyses about the RR estimate for treatment failure, but none had significant changes in the degree of heterogeneity among the included studies. Moreover, we excluded outliers [35, 40] and/or

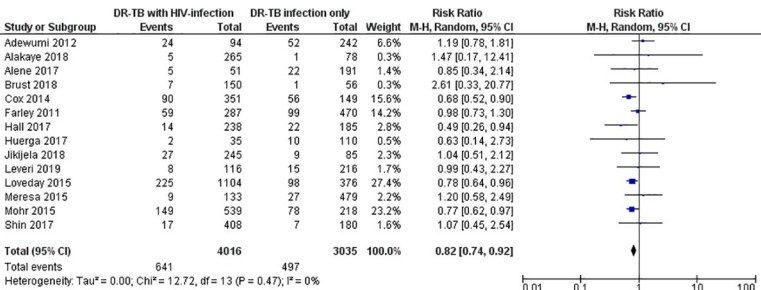

**Fig 5. Forest plot for risk of loss from treatment among HIV-infected versus HIV-unifected patients treated by second-line tuberculosis therapy.**

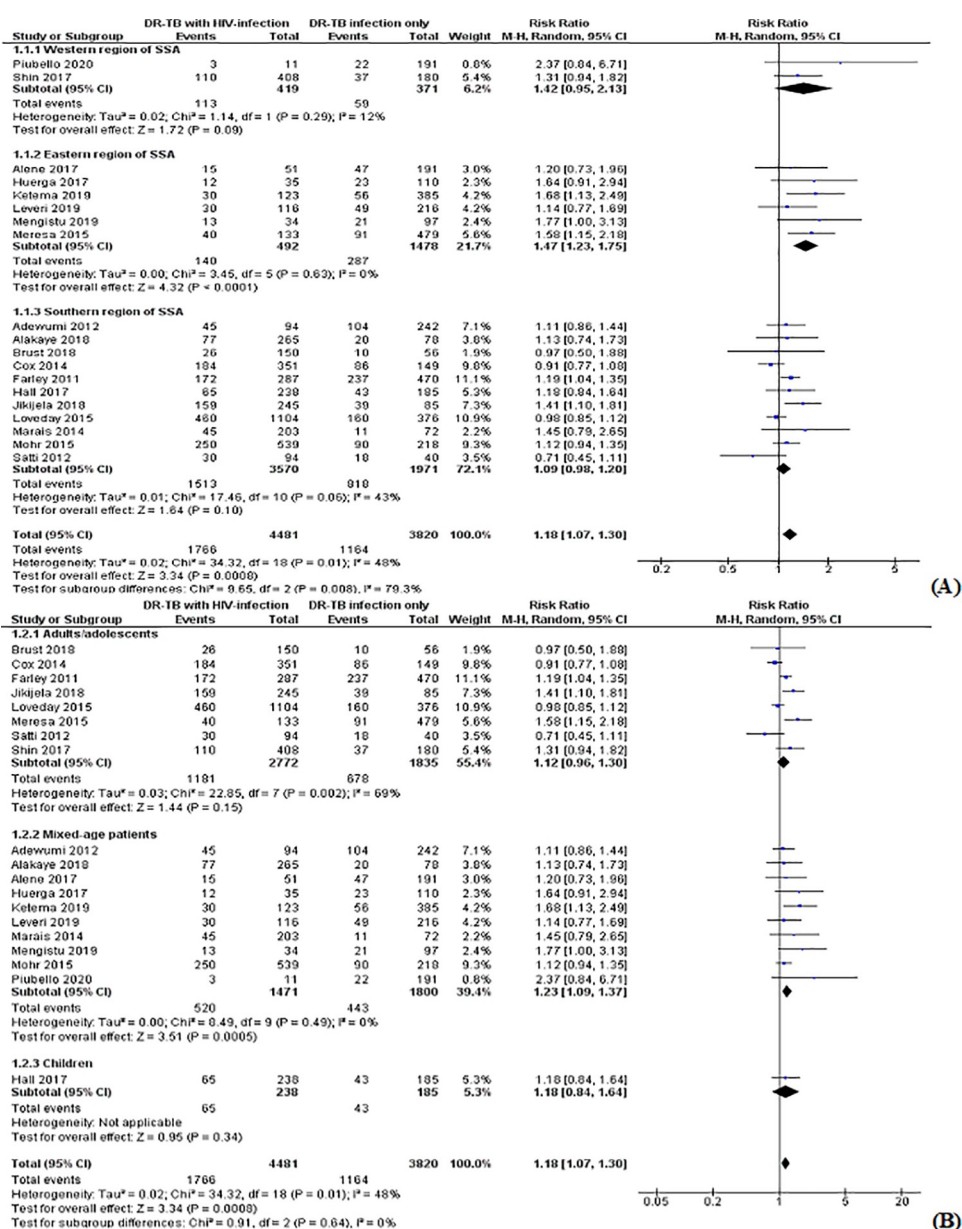

**Fig 6. Forest plot depicting subgroup analyses for the risk of unfavorable treatment outcome among HIV-infected versus HIV-uninfected patients treated with second-line TB therapy.** (A) Patient age groups, and (B) Regions of SSA.

more studies [27, 29], but they did not have significant changes in the degree of heterogeneity among the included studies. As a result, we included all the studies reporting deaths (n = 16), treatment failure (n = 10) and loss from treatment (n = 14) in their respective meta-analyses.

## Publication bias

We assessed the effects of small-studies (publication bias) on our estimates using funnel plots under the fixed-effects model that helped us visualize the symmetry status of each funnel plot (i.e., presence of symmetrically inverted funnel in the absence of bias). However, we did not perform either Egger's regression test or Begg's correlation test as two of the required criteria

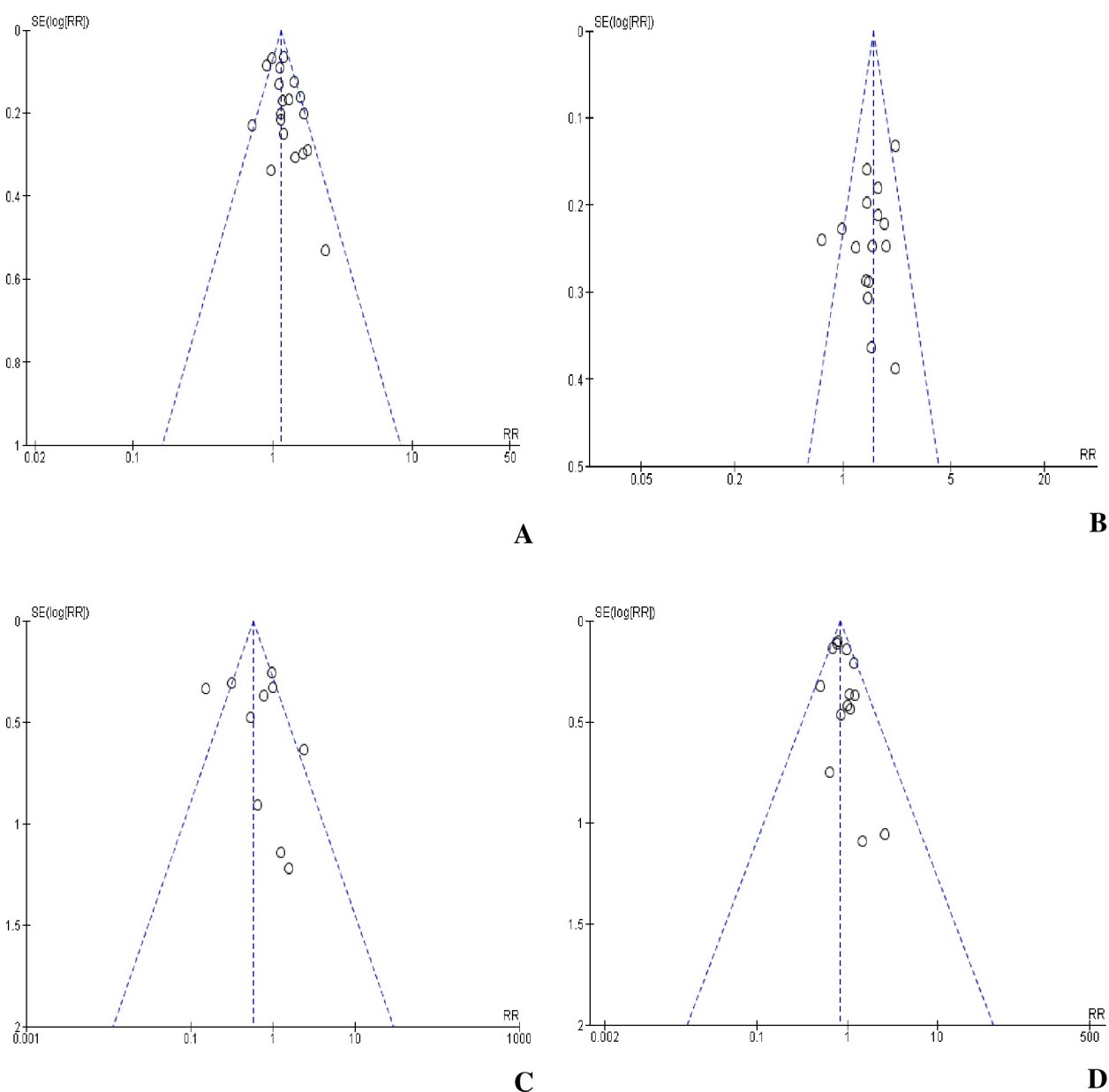

**Fig 7. Funnel plot of standard error by logit RR for publication bias.** (A) Overall unfavorable treatment outcome; (B) Deaths; (C) Treatment failures, and (D) Losses from treatment.

for appropriateness of these quantitative tests were not fulfilled. That means, the ratio of extreme variance across studies was not greater than four and the pooled RR estimate for the unfavorable outcome had a statistically significant heterogeneity ($I^2 = 48\%$; $P = 0.01$) even though the studies included were $> 10$ in number and had more than one studies with significant results [42, 43]. In principle, visual evaluation for effect estimates from larger studies that spread narrowing at the top of the plot, with more widely scattered estimates at the bottom of the plot among smaller studies could inform the presence of bias. Again, the inclusion of unpublished studies in our analysis might have impacted and reduced of the risk of publication bias. Accordingly, none of the funnel plots figured out had effect estimates which were scattered more widely at the bottom of the plots (Fig 7A–7D).

## Discussion

The overall risk of unfavorable treatment outcome to second-line anti-tuberculosis drugs was 1.18 times higher among HIV-infected patients than in those HIV-uninfected ones. This was a significant increment by 18% in HIV-infected patients compared with the HIV-uninfected ones, both of them treated by second-line anti-tuberculosis medicines. Despite their varying magnitudes, several studies revealed higher rates of reports for an unfavorable outcome to second-line TB therapy in HIV-infected patients than in those with no HIV-infection. Accordingly, a study comparing the proportion of successful outcomes during the second-line TB therapy indicated that 28% of HIV-infected and 16% of HIV-uninfected patients had unfavorable outcomes [44]. Another study also explained that the risk of unfavorable outcome was 1.14 times higher among HIV-infected patients than in those HIV-uninfected [45]. Again, for several studies with different sample sizes but assessed treatment outcomes to second-line TB therapy, the risks of unfavorable outcome were 1.5 (in a study with 439 participants) [46]; 2.2 (in a study with 2,185 participants) [47]; 2.94 (in a study with 2,266 participants) [11]; 3.3 (in a study with 3,729 participants) [48]; 3.44 (in a study with 1,809 participants) [49]; 7.14 (in a study with 302 participants) [50]; 10.07 (in a study with 360 participants) [51]; 10.16 (in a study with 235 participants) [52]; and 41 (in 51 cases as participants) [53] times higher among HIV-infected patients than those HIV-uninfected ones. Alternatively, the likelihood of success for second-line anti-tuberculosis treatment was 2.3 times higher among HIV-uninfected patients than the HIV-infected ones [54].

Importantly, a lower rate of favorable outcome (48%) than a global rate of a successful outcome (54%) during the second-line TB therapy among HIV-infected patients is indicative of a difficulty to attain greater rates of favorable outcome among the TB/HIV-coinfected patients [55, 56]. In line with this, TB treatment interruptions and more hospital readmissions due to adverse drug events linked to interactions and treatment complexities might also contribute to the increased rate of unfavorable outcomes in the TB/HIV-coinfection [57, 58]. Also, negative influences such as social discrimination and inattention were important barriers to the effective treatment of the TB/HIV-coinfection with high mortality rates (i.e., death as a component of unfavorable outcome) [59]. Again, drug-interactions, overlapping toxicities, and inflammatory immune reconstitution syndromes are the unique treatment challenges and the cursed duets of TB/HIV-coinfection [60, 61]. Specifically, severe side effects and high-level fatigue, stress, and burden of stigma were critical barriers to patient adherence in MDR-TB/HIV-coinfection [62]. More remarkably, there is a hypothesis that explains differences in a variety of transcriptional patterns and expression of genes coded by the TB/HIV interactions resulting in increased inflammatory conditions that contribute to the unfavorable outcome [63].

Significantly increased risks of an unfavorable outcome to second-line anti-tuberculosis treatment among HIV-infected versus HIV-uninfected patients were revealed for some patient groups (i.e., 1.22 times higher in mixed-age patients and 1.47 times higher in the eastern region of SSA, compared to their respective counterparts), but the increases in rest of the subgroups were not significant. Consistently, reports of previous studies involving mixed-age patients indicated that older HIV-infected patients had 1.53 [64] and 4.8 [49] times higher hazards of unfavorable outcomes than those older HIV-uninfected. Again, pediatric and elderly individuals were among the mixed-age patients in whom immune statuses might not be competent compared with that of adults/adolescents counterparts. This could be related with immature and reduced immune functions in children and elderly individuals, repetively. However, treatment successes for both DR-TB and HIV-infections require optimal functions of immune cells. The immune functions of such individuals are further weakened in the presence of the HIV and TB co-infections compared with the TB infection alone. The increased risks of unfavorable outcomes in mixed-age patients treated with second-line TB therapy among HIV-

infected versus HIV-uninfected patients might hint these differences linked with worsening impacts of both infections on immune competence and age-related physiologic immaturity and/or changes. Alternatively, a study report also highlighted that there was no significant association of the unfavorable outcome by different age groups which aligns with the non-significant risk ratio in adolescent/adult groups treated by the second-line anti-tuberculosis drugs [65]. Again, lack of efficiently integrated services for HIV and DR-TB units in the eastern region of SSA could contribute to the higher risks of an unfavorable outcome to the second-line anti-tuberculosis treatment among the HIV-infected than those HIV-uninfected patients. In alignment with this finding, positive influences of interventions that effectively integrated programs of TB/HIV control strategies on outcomes of the two infections were indicated in the southern region of SSA [48, 66, 67].

Similarly, the risk of death during treatment with second-line anti-tuberculosis drugs was significantly increased by 50% (1.50 times higher) among HIV-infected patients compared with those HIV-uninfected ones. Consistently, variable but positive relationships indicated between mortality during course of the second-line anti-tuberculosis regimen and presence of HIV-infection with 1.46 (in a study with 3,802 participants) [68]; 1.7 (for ART initiated) and 2.3 (for no ART) (in a study with 3,566 participants) [69]; 2.35 (in a study with 147 participants) [70]; 4.2 (in a study with 1,768 participants) [71]; 5.6 (in a study with 2,097 participants) [72]; 5.6 (in a study with 1,209 participants) [73]; and 29.9 (in a study with 50 participants) [74] times higher risks or odds of death in the HIV-infected patients than in those HIV-uninfected counterparts. Again, reports for the rates of mortality during the second-line TB therapy among HIV-infected versus HIV-uninfected patients showed 14% versus 6% in a study with 206 participants [75], 20% versus 9% in a study with 671 participants [76], and 72% versus 20% in a study with 173 participants [77]; all of them aligned with our study finding. Besides, early mortality and mortality adjusted after default were the most common reasons justified for the higher risks of unfavorable outcomes during the second-line TB treatment in the HIV-infected versus HIV-uninfected patients [78, 79].

Different from the risks of death and overall unfavorable outcome, rates for the loss from treatment and treatment failure were relatively lower during the courses of second-line TB treatment in HIV-infected patients than in those HIV-uninfected. A non-significant decrease in the pooled risk of loss from treatment estimated was 0.82 times lower in HIV-infected patients than in those HIV-uninfected. In agreement to this finding, study reports indicated 11.8% versus 26.2% (P<0.001) and 14.2% versus 35.2% rates of loss to follow-up among the HIV-infected versus HIV-uninfected patients; and the loss to follow-up was 7.67 (95% CI: 1.00–59.0) times more likely in the HIV-uninfected patients than in those HIV-infected [80–82]. Also, reminding information via cell phone as a part of the HIV/TB integrated program enabled tracing lost patients and resulted in twice more likely returnee for traced patients than untracked ones [83]. Such a program could reduce the number of losses to follow-up in HIV/TB co-infection. Again, a relatively higher number of treatment defaults reported as a death in HIV-infected versus HIV-uninfected MDR-TB patients through continuous tracing of patients lost from follow-up could indicate its negative impact on the number of loss from treatment among the HIV-infected patients [78]. Besides, patient-provider interactions were likely stronger for more patient empowerment and support during second-line TB therapy among HIV-infected patients compared with those HIV-uninfected [84, 85]. More importantly, higher rates of severe adverse drug events and hospital readmissions due to the events during second-line TB treatment in HIV-infected patients than in those HIV-uninfected might reduce the rates of loss from treatment [86, 87].

Despite inconsistencies in reports of some studies regarding this finding, infection management strategies for both HIV and TB programs could have a synergistic effect that can reduce

the risks of loss from treatment during periods of the second-line TB therapy in patients with HIV-infection. We estimated a statistically non-significant reduced risk of treatment failure in HIV-infected patients than in those HIV-uninfected. In line with this, a study report also highlighted a non-significant increase in odds (1.1 times) of the treatment failure among HIV-infected patients [73]. Alternatively, a study report indicated 1.6 times higher odds of treatment failure in HIV-uninfected patients than in those HIV-infected [79]. Again, a previous study explained more frequent support for compliance in HIV-infected patients (23%) than in those HIV-uninfected (7%) [81]. Accordingly, good compliance with treatment is a key component of strategies that can reduce the rates of treatment failure [88–94].

Despite the large individual patient data pooled for this meta-analysis, it is not without limitations. First, the studies considered for this meta-analysis were observational by nature. This selection might have resulted in a higher degree of heterogeneity with a range of potential biases. However, we employed a random-effects model of analysis which is an appropriate method in such an anticipated heterogeneity. Besides, we executed sensitivity and subgroup analyses to reduce the degree of heterogeneity. Second, we included articles written in the English language and this could under-or over-estimate the pooled RR estimates for unfavorable outcomes during second-line TB therapy in the SSA. Therefore, interpretations of these findings should be seen in context of the aforementioned limitations.

## Conclusions

We found that the risk of overall unfavorable outcome to second-line TB therapy among patients treated in SSA was significantly higher in HIV-infected patients compared with those HIV-uninfected. It was highly increased in mixed-age patients and the eastern region of SSA. The risk of death was alarmingly increased by 50%, but both treatment failure and loss from treatment were the outcomes with decreased rates in the HIV-infected patients compared with those HIV-uninfected. Therefore, special strategies that reduce the risks of death should be discovered and implemented for HIV and DR-TB co-infected patients on second-line tuberculosis therapy. Besides, integrating the HIV and DR-TB treatment strategies in the eastern region of SSA could optimize outcomes of HIV-infected patients during their second-line TB therapy.

## Supporting information

**S1 Table. Quality assessment for the included studies in meta-analysis.**
(DOCX)

**S2 Table. Completed PRISMA 2009 checklist.**
(DOC)

**S1 Fig. Forest plot for the risk of death among HIV-infected versus HIV-uninfected patients treated with second-line TB therapy subgrouped by regions of the SSA.**
(TIF)

**S1 File. Abbreviations (acronyms).**
(DOCX)

## Acknowledgments

We would like to thank Tadesse Bekele (PhD) and other staff of the Haramaya University, College of Health and Medical Sciences without them this meta-analysis would not have been realized.

## Author Contributions

**Conceptualization:** Dumessa Edessa.

**Data curation:** Dumessa Edessa, Mekonnen Sisay, Yadeta Dessie.

**Formal analysis:** Dumessa Edessa, Mekonnen Sisay, Yadeta Dessie.

**Investigation:** Dumessa Edessa, Mekonnen Sisay, Yadeta Dessie.

**Methodology:** Dumessa Edessa, Mekonnen Sisay, Yadeta Dessie.

**Software:** Dumessa Edessa.

**Supervision:** Dumessa Edessa, Mekonnen Sisay, Yadeta Dessie.

**Writing – original draft:** Dumessa Edessa.

**Writing – review & editing:** Dumessa Edessa, Mekonnen Sisay, Yadeta Dessie.

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
