## [Decision Letter · Decision Letter 0]

17 Jun 2020

PONE-D-20-11009

Unfavorable outcomes to second-line tuberculosis therapy among HIV-infected versus HIV-uninfected patients in sub-Saharan Africa: a meta-analysis of individual patient data

PLOS ONE

Dear Dr. Dumessa Edessa, M.Pharm

Your manuscript, entitled "Unfavorable outcomes to second-line tuberculosis therapy among HIV-infected versus HIV-uninfected patients in sub-Saharan Africa: a meta-analysis of individual patient data" has been subjected to a double-blind review process by two reviewers who are experts in the related fields. Enclosed please find the reports from these reviewers.

Based on the reviewers' recommendations, I am delighted to inform you that your manuscript has been ACCEPTED WITH MINOR REVISIONS for the PLoS ONE.

Please note that it is imperative for you to revise the manuscript according to reviewers' comments and guidelines. Please use the TRACK CHANGES feature of MS-Word to make your revisions. To do so, please select the TRACK CHANGES option in the TOOLS menu. Choose HIGHLIGHT CHANGES then 'tick' the box next to TRACK CHANGES WHILE EDITING by one left click in the box, then choose OKAY. When you delete or add text, the 'TRACK CHANGES' then shows the changes in color and crosses out automatically your deletions with a color line leaving them highlighted so we can identify where your corrections have been made. Once you have revised the manuscript, please re-submit your new version to the PLoS ONE on or before 09/16/2020, with a cover letter outlining point-by-point the revisions you have made in regards to the reviewers' comments and guidelines.

Thank you very much for submitting your article to the PLoS ONE. I look forward to receiving the revised version of your manuscript.

Sincerely,

Andy Yu, MD/Ph.D

Academic editor

2. Please include a discussion of how/whether you searched for unpublished studies (e.g. grey literature) in order to reduce the impact of publication bias on your results.

Reviewers' comments:

Reviewer's Responses to Questions

**Comments to the Author**

1. Is the manuscript technically sound, and do the data support the conclusions?

Reviewer #1: Yes

Reviewer #2: Yes

2. Has the statistical analysis been performed appropriately and rigorously? 

Reviewer #1: Yes

Reviewer #2: Yes

3. Have the authors made all data underlying the findings in their manuscript fully available?

Reviewer #1: Yes

Reviewer #2: Yes

4. Is the manuscript presented in an intelligible fashion and written in standard English?

Reviewer #1: Yes

Reviewer #2: Yes

5. Review Comments to the Author

Reviewer #1: Drug-resistant tuberculosis (DR-TB), including multidrug-resistant TB (MDR-TB) and extensively drug-resistant TB (XDR-TB), is considered a potential obstacle for elimination of TB globally. HIV coinfection with DR-TB further complicates the scenario. HIV-1 comanagement with ART could worsen outcomes of second-line anti-tuberculosis drugs.

In this study, authors measured the risk ratios for the unfavorable outcomes in HIV-infected TB patients compared with HIV-uninfected TB patient by analyzing 19 studies. They found both a higher risk of an unfavorable TB treatment outcome and higher risk of death in HIV-infected patients. Their results indicate that special strategies that reduce the risk of death should be discovered and implemented to treat HIV-infected patients with second-line anti-TB therapy.

I recommend to accept this manuscript for publication.

Reviewer #2: In this systematic review article, the authors estimated the risks of unsuccessful outcomes (death, treatment failure, and loss from treatment) to second-line tuberculosis therapy in HIV-infected versus HIV-uninfected patients with drug-resistant tuberculosis pooled from 19 eligible studies. Meta-analysis was performed on 4481 HIV-infected and 3820 HIV uninfected African DR-TB patients, 1766 and 1164 patients having unfavorable outcomes, respectively. They found a higher risk ratio (RR) of overall unfavorable outcomes and death in HIV-infected patients. In addition, subgroup analysis showed a higher RR of unfavorable outcomes in mixed-age and in the eastern region of sub-Saharan Africa.

The systematic review and meta-analysis were done properly and there are no major concerns. Below are some minor concerns:

1) Did the DR-TB patients include MDR-TB patients?

2) Were all the HIV-infected TB patients on ART? How ART status might affect RR for death and overall unfavorable outcomes? Please list ART status in Table 1 and discuss how this could affect the RR for death and overall unfavorable outcomes.

3) When reporting overall significance, please use the exact p value instead of P>0.05 or <0.05.

4) Was the RR for death higher in the eastern region of SSA?

5) It is not clear what the significance of a higher RR in mixed age HIV-infected and uninfected patients. As RR in the adult and adolescent subgroup was not significantly different between HIV-infected and uninfected patients, is the difference in the RR for the overall unsuccessful outcomes mainly from pediatric patients?

6) In sensitivity and subgroup analyses, please provide description of outliers

7) The overall z score and p value need to be added to the Sensitivity and subgroup analyses section.

8) The figures are of very low quality and are illegible.

6. PLOS authors have the option to publish the peer review history of their article (what does this mean?). If published, this will include your full peer review and any attached files.

Reviewer #1: No

Reviewer #2: No

---

## [Author Response · Author response to Decision Letter 0]

22 Jul 2020

Dears,

Our response to reviewers' comments is attached with the manuscript document through submission process. Please look into it.

---

## [Editor Report · Decision Letter 1]

29 Jul 2020

Unfavorable outcomes to second-line tuberculosis therapy among HIV-infected versus HIV-uninfected patients in sub-Saharan Africa: a systematic review and meta-analysis

PONE-D-20-11009R1

Dear Dr. Dumessa Edessa, M. Pharm,

We’re pleased to inform you that your manuscript has been judged scientifically suitable for publication and will be formally accepted for publication once it meets all outstanding technical requirements.

Kind regards,

Qigui Yu, M.D./Ph.D

Academic Editor

PLOS ONE

Additional Editor Comments (optional):

Dear Dr. Dumessa Edessa, M. Pharm,

It's my pleasure to inform you that, after the peer review, your manuscript entitled "Unfavorable outcomes to second-line tuberculosis therapy among HIV-infected versus HIV-uninfected patients in sub-Saharan Africa: a meta-analysis of individual patient data" has been ACCEPTED for the PLoS ONE. 

Sincerely,

Andy Yu, MD/Ph.D

Academic editor